# Development of the Polish Version of the ICF Core Set for the Environment of Older People

**DOI:** 10.3390/ijerph192316341

**Published:** 2022-12-06

**Authors:** Agnieszka Ćwirlej-Sozańska, Bernard Sozański, Anna Wilmowska-Pietruszyńska, Beata Kizowska-Lepiejza, Agnieszka Wiśniowska-Szurlej

**Affiliations:** 1Institute of Health Sciences, College of Medical Sciences, University of Rzeszow, Rejtana Street 16C, 35-959 Rzeszow, Poland; 2Laboratory of Geronto-prophylaxis, Center for Innovative Research in Medical and Natural Sciences, Rzeszow of University, Warzywna Street 1A, 35-310 Rzeszow, Poland; 3Institute of Medical Sciences, College of Medical Sciences, University of Rzeszow, Rejtana Street 16C, 35-959 Rzeszow, Poland; 4Faculty of Medicine, Lazarski University, Świeradowska Street 43, 02-662 Warsaw, Poland; 5Center for Foreign Language Studies, University of Rzeszow, Rejtana Street 16C, 35-959 Rzeszow, Poland

**Keywords:** ICF, environment, core set, aged

## Abstract

Introduction: The percentage of older people in Polish society increases every year. The interaction between the individual health condition and the barriers in the environment of the elderly leads to the development of disability and the limitation of activity and participation in daily activities. Aim: This study was aimed at selecting the category of the International Classification of Functioning, Disability, and Health (ICF) to assess the environment of older adults in Poland in the context of their daily functioning. Materials and methods: The study was designed to develop a user-friendly tool collecting ICF-based data on the living environment of older people, consisting of five phases: (1) the systematic review of the literature, (2) the empirical multicenter study, (3) the qualitative study based on interviews conducted among the elderly, (4) the experts’ study—an assessment of selected codes from the perspective of experts, (5) the consensus conference. Results: Consensus was reached for 20 ICF categories, creating a comprehensive core set for the assessment of the living environment of older people, which included six codes from chapter 1, *Products and technology*, three codes from chapter 2, *Natural environment and human-made changes to the environment*, four codes from chapter 3, *Support and relationships*, four codes from Chapter 4, *Attitudes*, and three codes from Chapter 5, *Services, systems, and policies*. Conclusions: The core set for the assessment of the living environment of older people living in Poland is a comprehensive and important set of 20 ICF codes that reflect the most important elements of the environment affecting the health and functioning of the elderly. This set can contribute to the optimal management of care services and support in the area of adapting the environment to the older population. The core set for environmental assessment was developed for use by medical and care facilities, as well as by social workers, who should also pay attention to the elements of the environment that affect the level of functioning of older people. In the future, it may also form the basis of national surveys and screening tests for the assessment of the living environment of older people. Optimizing and enhancing the surrounding environment can contribute to a greater degree of independence, even with existing health problems in the older population.

## 1. Introduction

The proportion of older people in Polish society increases every year. In 2019, the percentage of people aged 65 years or over reached 18.1% of the Polish population [1]. Furthermore, the prevalence of health problems increases with age [2]. With reference to the older population, it is worth mentioning the characteristic reduction in muscle mass and strength [3,4], as well as the impairment of mobility and balance [5]. Moreover, the incidence of cognitive disorders is rising [6]. The number of chronic diseases is also increasing, especially diseases of the musculoskeletal system, cardiovascular and respiratory diseases, as well as visual and hearing impairment [7]. The interaction between the individual health condition and the barriers in the environment of older adults leads to the development of disability and the limitation of activity and participation in everyday activities. Activity limitations and participation restrictions lead to secondary physical (low physical activity) and mental changes (e.g., due to loneliness and limited interpersonal contacts) in the elderly [8,9]. The weakness syndrome develops, increasing the risk of disability and dependence [10,11], a low quality of life [12], higher costs of social and health care [13], hospitalization [14], institutionalization [15], and premature death [16].

Disability, however, is not only a health problem but is the result of an interaction between humans and factors in the environment [17]. If the living environment of an older person is accessible, attitudes and social norms are positive, and policies and services take into account all the needs of older people, then the phenomenon of disability is significantly reduced. In a favorable environment, being able to stay physically active and participatory slows the development of many diseases and prevents disability [18].

Interventions improving the environment enable people to remain independent and do the things that are essential. Owing to them, health and functioning are improved. Therefore, it is important to define what environmental barriers are present in the environment of an older person and consider their impact on the level of his/her disability. Adequate and effective interventions can only be achieved if disability is able to be assessed in the context of the subject’s environment [19].

In order to understand the interaction between individuals’ health states and the contexts in which they live, a detailed analysis of the various elements of that interaction is worth considering. The International Classification of Functioning, Disability, and Health (ICF) provides a framework for the description and analysis of this interaction and thus for understanding the impact of the environment on human functioning.

The fundamental aspect of ICF is its universal character. This means that any person with a particular state of health can be described using this classification. The ICF takes an integrated approach that recognizes the influence of both individual characteristics and environmental factors that should be considered in the process of disability assessment. Disability is a complex and multidimensional phenomenon. It often requires complex environmental conditions, including strong social support, to reduce it. Moreover, the environment of each person is different and may have a different influence on the exacerbation of disability. The ICF emphasizes the integral role of the environment in human functioning. As the living environment is very diverse, there are problems in assessing it. To our knowledge, unfortunately, there is no ICF-based tool for assessing the living environment of older adults.

The aim of the study was to select codes and develop an ICF-based tool for assessing the environment of older adults living in Poland in the context of their functioning and disability. In this article, we present the stages of code selection and the results of the consensus reached on the description of the environment of older people.

## 2. Materials and Methods

This study was designed in accordance with WHO ICF Research Branch guidelines [20] to develop a user-friendly tool for collecting data on the living environment of older people for use in the clinical and social support process of older people based on ICF in Poland.


*ICF Environmental Factors*


The environmental factors in ICF describe the physical and social “context” in which a person lives, works, and participates. In ICF, environmental factors are termed as “facilitating factors” or “barriers” depending on the nature of their interaction and the resulting health experience [21]. Environmental factors in the classification are placed on two levels: individual—the immediate personal environment of a person, and social—social structures, services, and systems. Environmental factors interact with the functioning and disability components of ICF. The basic idea of environmental factors is a facilitating or hindering influence of the physical world, the social world, and the system of attitudes. ICF identifies and maps factors that contribute to disability but also those that facilitate health and participation. In this respect, the structure of ICF is sensitive to changes in experiences over time in the continuum of health disabilities and in different environments [22]. However, in Poland, an environmental assessment of life in the context of the disability of the elderly based on ICF has not been developed and applied so far.

Environmental factors are quantified using the same general ICF scale as the other components of the classification. In the case of environmental factors, the first qualifier can be used both to define the range of positive aspects of the environment, i.e., facilitators, and to define the range of negative effects, i.e., barriers. The same scale, 0-4, is used in both cases, but the decimal point is replaced with a plus sign [21] for facilitators.


*The Process of Developing a Basic ICF Set for Assessing the Living Environment of the Elderly*


The tool preparation process was carried out in multi-level analysis in five stages:(1)The Systematic Review

The first phase was the systematic review of the literature to identify environmental factors posing obstacles and barriers as well as to facilitate the functioning of older people aged 65 and over. The study was conducted in accordance with the methodology indicated by the WHO ICF Research Branch [20]. The following databases were searched: PubMed; MEDLINE; Google Scholar; and ISI Web of Knowledge. The following MeSH headings were used: Aged; 80 and over; Activities of Daily Living; Walking; Residence Characteristics; Environment. The qualification of publications was based on an analysis of the title, the abstract, and then the full-text version. The following issues were taken into account: the reviewed articles in English; articles assessing the living environment of the elderly; randomized controlled trials; clinical controlled trials; cross-sectional studies; observational studies; and qualitative studies. Searching for selected MeSH in medical databases provided 2988 records, and after subsequently removing duplicates and items not related to the subject after an initial analysis of abstracts, a total of 534 references remained. Then, publications that did not have a full text were removed, resulting in 517 articles. Ultimately, 97 articles were included in the analysis, which were used to collect the concepts of the positive and negative impacts of the environment on the functioning and disability of the elderly. The issues gathered from the literature review were linked with ICF categories using standard linking rules [23]. An illustration of how concepts are related to ICF categories is shown in Figure 1.

(2)The Empirical Multicenter Study

The second phase was the empirical multicenter study, which aimed to identify the problems experienced by older adults because of environmental barriers that are documented in institutional settings. In the first stage of this study, a database of centers and organizations cooperating or caring for the elderly was collected. The inclusion criteria for the center were at least 3 years of operation in the market, a profile of medical, social, caring, supporting, or mixed activities addressed to the older people, consent of the center, and willingness to participate in the study. The second stage was the selection of five medical centers and five centers of mixed or supporting profiles for the elderly from the gathered database. If a given center refused to cooperate or did not have an expert with at least 3 years of work experience, another center was drawn from the pool.

During the empirical multicenter study, employees of the institution participating in the study conducted their standard measurements and examinations with the beneficiaries, extended by a deep, semi-structured interview focusing on problems in daily functioning in the context of the environment and living conditions of the subjects. The instrument used to collect the data was the ICF checklist enriched with environmental codes selected after a systematic review of the literature. The ICF checklist required the researchers to evaluate the extent of the problem in each ICF category and the size of the problem/facilitator in terms of environmental factors. In addition, it was asked to focus on the relationship between the categories of functioning and activities with environmental categories and to mark those that had a substantial impact on the improvement or deterioration of the performance of activities. A category that has been identified as a problem, barrier, and/or facilitator for minimum of 25% of the subjects was included in the list of candidate categories.

(3)The Qualitative Study

The third phase was the qualitative study, which aimed to recognize environmental factors that have a significant impact on the functioning of older people from their perspective. The study was conducted based on consolidated criteria for reporting qualitative research (COREQ) and the WHO ICF Research Branch guidelines [20]. A detailed description of this study is provided below.


*Characteristics of the Research Team*


The main researcher, an experienced scientist in the fields of physiotherapy, geriatrics, geronto-prophylaxis, and public health with the title of Associate Professor, was responsible for the organization of the research team, which included four people in total with a minimum PhD title and research experience similar to that of the main researcher. All persons were professionally involved in the treatment, rehabilitation, education, and care of older people, and they were instructed in the area of conducting qualitative research in accordance with the COREQ guidelines. There were four women and one man in the research team. The research team met regularly from April to May 2022 for weekly meetings during the data collection phase.


*Qualitative Research Design*


The qualitative study was based on the method of the triangulation of research methods using semiotics, in-depth individual interviews, and focus group interviews in the research project. This made it possible to obtain a broader research material and a more complete basis for its interpretation. The study was carried out based on three focus groups including seven people each and nine semi-structured interviews with the participants of the study in order to interpret their experiences and opinions related to the functioning of older people in the broadly understood living environment. The research team applied the maximum diversification strategy to the focus group. A moderator and an assistant participated in each focus group session. The role of the moderator was to ask questions according to the established protocol (guide).

With references to nine older people, the study was performed as a semi-structured one-on-one interview by researchers. The interviews were conducted among people who, due to health or organizational reasons, could not participate in the focus group.

Both focus groups and face-to-face interviews were recorded using a dictaphone with the consent of the study participants and then transcribed (transcription process) for thematic analysis and presentation of the results.

Qualitative research was carried out in the period from April to May 2022. For the interviews, the participants were invited to the Laboratory of Geronto-prophylaxis at the Center for Innovative Research in Medical and Natural Sciences of Rzeszow University, whereas face-to-face interviews were conducted at the subject’s place of residence.


*Study Participants*


Study participants were recruited by means of a variety of methods, including targeted sampling and the distribution of a recruitment flyer. Additionally, a snowball sampling technique was used, asking study participants, at the end of each completed interview, to invite acquaintances who might be interested in participating in the project. The potential participants were informed about the purpose of the study.

The inclusion criteria for qualifying participants for the study were: age of at least 65 years, living in a society, verbal contact with the subject, cognitive state enabling the interview (AMTS—Abbreviated Mental Test Score > 6 points), and informed consent to participate in the study. It was assumed that at least half of the respondents will have at least one limit in IADL or ADL. The group structure reflected the social structure in terms of gender and place of residence. The exclusion criteria were: age under 65 years and current stay in a nursing home or hospital. The adopted criteria enabled the team to recruit a diverse sample of study participants.

A minimum research sample of 30 people was presupposed. Then, 140 older people appeared with the invitation to the study, and, finally, 84 people expressed their willingness to participate in the study. Based on the completed forms, the subjects were randomly selected in order to select a group of 30 people meeting the assumed criteria. Table 1 shows the structure of the study group in the qualitative research.


*Data Collection and Instrument*


The interview protocol (guide) was developed by the main researcher and then reviewed by a team member with the title of Professor and many years of experience in research and professional work with older and disabled people. The interview guide was then assessed and approved by all members of the research team. The guide was prepared in the form of a checklist arranged according to the environmental areas that were identified and linked to specific ICF categories during the literature exploration and multicenter study. At the end of the interview, each participant could add their comments and opinions on other elements of the living environment that were important in their opinion and were not raised during the meeting. The guide was validated in a pilot study with three older people.

(4)The Experts’ Study

The fourth phase was the experts’ study, which involved interviews among experts. The expert’s study was conducted in the form of an internet questionnaire and was aimed at gathering opinions and comments on the aspects of the functioning of the elderly in the context of environmental factors. The study was carried out on the basis of a protocol prepared in the form of a checklist arranged in accordance with the environmental areas that were defined and linked to specific ICF categories during the literature research, research in centers, and qualitative research.

The criterion for including experts was at least 5 years of professional experience in the functioning, disability, and health of the elderly. The experts were selected in two stages. In the first stage, a database of experts was created by means of contact with professional organizations and societies and authors of publications in each field, as well as using informal social and professional networks in Poland. In that way, a database of experts has been gathered, providing a group of people who are physicians, physiotherapists, occupational therapists, nurses, and people working in the field of care and social assistance for the elderly. The second step was to draw experts in each discipline from this pool. If an expert refused to participate in the study, another one was drawn from the pool.

The expert survey consisted of open-ended questions like those asked in the qualitative research. The experts’ responses were identified and then broken down and linked with the ICF. The category was only counted once for each expert. It was assumed that ICF categories that would be considered very important or were reported by at least three experts were included in the list of categories that qualified for the next stage.

The final list of categories to be assessed at the consensus conference consisted of codes that were considered valid at at least three stages of the research.

(5)Conference to Reach a Consensus

The fifth phase was the consensus conference, which contributed to the confirmation of the categories selected for the Polish environment core set and to the development of simple, intuitive descriptions of selected ICF categories. In that form, they are unambiguous and clear for older adults. Moreover, the fifth phase resulted in providing sample questions for the assessment of individual categories.

The consensus process involved three groups, each with seven experts. Experts from all over Poland with a minimum of 5 years of experience in caring for the elderly and/or conducting research in this area were invited to participate in the conference. They represented various areas of specialization related to work with the elderly. Each working group reflected a comparable representation of occupations/disciplines, areas of Poland, and genders. One expert was appointed to be the moderator for each group. Participants remained with their group throughout the consensus process. The language spoken at the conference was Polish. Each group received initial proposals of simple, intuitive descriptions of 20 selected ICF categories and proposed questions to be asked to the examined person. Descriptions were prepared on the basis of the results of similar conferences in other countries [24,25,26].

Participants were asked to review the proposed code set and to read and discuss the initial proposed descriptions and questions to be asked to the subject to assess the qualifier. The proposed set of codes was approved by a majority of over 75% of the votes. Participants voted on each description regarding whether each description was simple and intuitive enough and at the same time faithfully reflected the content of the code and whether the sample question allowed the subject to obtain answers to the content of the code. The first vote was to reach a consensus if the description reached 75% or more agreement in each working group. Following the presentation of the results and the discussion in the plenary session, the categories that did not reach a consensus in the first vote were split among the three working groups, and each group was asked to propose a new description for each category allocated. In the next plenary session, each proposal from the second session of the working group was discussed and put to a vote. As in the first vote, a consensus on the description was reached when at least 75% of all participants agreed that the new description was simple, intuitive, and faithful to the content of the code. At the third and final stage of the consensus conference, each working group was asked to develop a new proposal for each of the ICF categories, which was still not accepted after the second vote. At the third and final plenary session, each participant was asked to vote for one of the three descriptions.

## 3. Results


*The Systematic Review*


Concepts collected from the literature review were linked with a total of 35 ICF categories. With respect to ICF environmental factors, eight codes were selected from chapter 1, *Products and technology*, six codes were selected from chapter 2, *Natural environment and human-made changes to the environment*, eight codes were selected from chapter 3, *Support and relationships*, nine codes were selected from chapter 4, *Attitudes*, and five codes were selected from Chapter 5, *Services, systems, and policies* (Table 2).


*The Empirical Multicenter Study*


Concepts collected during a visit to 10 different centers and organizations cooperating with or dealing with older people were linked to a total of 23 ICF categories and included seven codes from chapter 1, *Products and technology*, three codes from chapter 2, *Natural environment and human-made changes to the environment*, five codes from chapter 3, *Support and relationships*, six codes from chapter 4, *Attitudes*, and three codes from chapter 5, *Services, systems, and policies* (Table 2).


*The Qualitative Study*


Concepts collected during focus groups and interviews from 30 randomly selected older people living in the community were linked to a total of 20 ICF categories and included six codes from chapter 1, *Products and technology*, three codes from chapter 2, *Natural environment and human-made changes to the environment*, four codes from chapter 3, *Support and relationships*, four codes from chapter 4, *Attitudes*, and three codes from chapter 5, *Services, systems, and policies* (Table 2).


*The Experts’ Study*


Concepts collected from 12 experts were linked to a total of 22 ICF categories and included six codes from chapter 1, *Products and technology*, three codes from chapter 2, *Natural environment and human-made changes to the environment*, four codes from chapter 3, *Support and relationships*, five codes from chapter 4, *Attitudes*, and four codes from chapter 5, *Services, systems, and policies* (Table 2).


*A Set for the Assessment of the Living Environment of Older People in Poland*


On the basis of a matrix made of codes linked to the concepts obtained in phases 1–4, an initial core set was created to assess the living environment of older people living in Poland. The set included categories that overlapped at at least in three phases of the study. Finally, a set of 20 ICF codes was received, including six codes from chapter 1, *Products and technology*, three codes from chapter 2, *Natural environment and human-made changes to the environment*, four codes from chapter 3, *Support and relationships*, four codes from chapter 4, *Attitudes*, and three codes from chapter 5, *Services, systems, and policies* (Table 1).


*Conference to Reach a Consensus*


The 20 ICF categories selected in the research process were validated during the consensus conference. The final, simple, intuitive descriptions and sample questions for each category of the ICF Basic Set are determined by a majority of votes and are presented in Table 3.

## 4. Discussion

The relationship between the level of disability and the arrangement of the living environment of older adults has been proven [27]. Disability is not only a health problem but is the result of an interaction between a person with a certain health condition and factors in his/her life environment [17]. Therefore, when organizing research on the disability of the elderly, the role of environmental factors cannot be ignored. Only a complete, comprehensive picture of the life situation of older people allows for a proper assessment of the factors increasing or facilitating difficulties in the functioning of older adults. It also allows researchers to plan supporting activities, preventing the development of functional limitations, thus delaying the disability and dependence of the elderly and improving their quality of life. There are a number of elements to consider in order to understand disability [28]. These elements are the person, his/her health condition, and personal characteristics, i.e., personal factors, external context, or the environment that a specialist can describe and assess objectively, as well as the subject’s perception of the environment. Human–environment interactions are complex and multi-layered, as reflected in the ICF approach. The environment surrounding a person can be described in a hierarchical way, i.e., as an immediate one—the closest environment of the subject, such as the place of residence, family members, and access to healthcare workers—and the other one— the more general environment relating to, e.g., culture or political systems shaping the general realities of life in a particular country. Therefore, the purpose of a comprehensive assessment of an older person’s environment is to explain how different aspects of the immediate and general environment affect their disability.

With reference to social and clinical work, an immense value is the simulation of how various environmental factors affect an individual in a particular state of health and psyche, influencing the increase or reduction in his/her disability, the deterioration or improvement of functioning, and the quality of life [29,30]. It is worth mentioning that, due to this determination, it is possible to plan the comprehensive rehabilitation of an individual patient and adapt the immediate environment of an individual’s life to support the functions, as well as to shape policies and systems that ensure an accessible, beneficial, and health-promoting environment.

Taking everything into account, the assessment and analysis of environmental factors related to the disability of the elderly is of particular importance [31]. Adapting the living environment of older adults, removing barriers, and introducing facilitators can significantly reduce the costs of health and social care. This is especially important in the context of an increasing life expectancy and the rise in the percentage of older people in society [32].

The analysis of these factors should be country-specific. Moreover, it also seems necessary to build instruments allowing for an efficient and easy assessment of the environment. In our study, in the course of the process of selecting codes based on WHO recommendations, the 20 most important ICF categories in the field of environmental factors were finally identified. The categories selected include five ICF chapters. Six codes were selected from chapter 1, *Products and technology*, including: E110—Products or substances for personal consumption, E115—Products and technology for personal use in daily living, E120—Products and technology for personal indoor and outdoor mobility and transportation, E125—Products and technology for communication, E150—Design, construction, and building products and technology of buildings for public use, and E155—Design, construction, and building products and technology of buildings for private use.

Scientific evidence confirms that proper supplementation, hydration of the body, the correct selection of drugs, and the prevention of polypharmacy are extremely important in maintaining a good physical and mental condition of the elderly [33,34]. A proper, balanced diet, adapted to the health condition and needs of older adults, ensures proper nutrition of their body, delaying or preventing pathological changes in human systems and organs [35]. Adequate assistive and medical devices and other personal aids allow an individual to maintain the comfort associated with everyday activities [36,37]. A properly organized home environment can maintain or improve people’s physical and mental health, ensure their well-being, and enable them to carry out their daily activities safely and comfortably. Installing aids and adaptations in the homes of the elderly can improve the accessibility and usefulness of a person’s home environment, maintaining or restoring a person’s ability to perform daily activities [32]. It is also very important to adapt public space and public utility facilities to the needs of the elderly [38]. Environmental barriers such as poor street conditions, high curbs, a lack of benches or pedestrian zones, etc. limit mobility [39]. Furthermore, other important barriers are also problems regarding access to transport and difficulties with access to health centers [40]. The withdrawal of older people moving around the space outside the place of residence or dealing with various matters causes limitations in physical capacity and cognitive functions [38]. Environmental barriers hindering outdoor mobility accelerate the decline of autonomy in external participation among older people living in the community [40].

In our study, we selected three codes from chapter 2, *Natural environment and human-made changes to environment*, including: E210—Physical geography, E240—Light, and E260—Air quality. Environmental barriers such as hills in the surrounding area, distance to services, uneven ground, snow and ice, poor lighting, and the lack of pedestrian zones impair the movement of older adults outside their place of residence [39]. Air pollution (smog) impairs respiratory functions, leads to diseases of the respiratory system, and increases the risk of death [41].

The next categories selected in our process were four codes from chapter 3, *Support and relationships*, including: E310—Immediate family, E325—Acquaintances, peers, colleagues, neighbors, and community members, E340—Personal care providers and personal assistants, and E355—Health professionals (professionals in healthcare). We also considered four codes from chapter 4, *Attitudes*, which are connected with the selected codes from chapter 3, including: E410—Individual attitudes of immediate family members, E425—Individual attitudes of acquaintances, peers, colleagues, neighbors, and community members, E440—Individual attitudes of personal care providers and personal assistants, and E450—Individual attitudes of health professionals (professionals in healthcare). Isolation and loneliness promote the development of depression and disability. Strengthening social contacts improves functioning and quality of life [42]. Some studies confirmed the effectiveness of programs that support the amount of social interaction and improve social skills [43]. Moreover, positive results of the created social networks supporting older people were also found [44]. Senior clubs, coffee mornings, and memory support care groups improve well-being and a sense of acceptance, increasing self-confidence and overall happiness [45]. Local initiatives, which are specific to the preferences of a given community, work well here [46]. Moreover, social assistance for older people with a significant impairment of body functions is extremely important in purchasing food and medicines, making an appointment to see the doctor, or helping to keep a warm and safe home [47]. Positive relationships with other people, both with family and acquaintances, are very important elements of the environment. Being able to receive help from others reduces barriers to the activity and participation of older people [48]. Maintaining social contacts is an important factor. The social participation of older adults is crucial for their active aging. It has a positive effect on physical and mental health and maintains capacity [49], cognitive functions [50], as well as a higher level of health-related quality of life [51]. Poor social relationships increase the risk of mortality [52].

The last set of categories selected as key categories for assessing the situation of older people covers the general background of the situation of older people. These are the three codes in chapter 5, *Services, systems, and policies*, including: E570—Social security services, systems, and policies, E575—General social support services, systems, and policies, and E580—Health services, systems, and policies. The availability of and trust in the healthcare system are very important in prophylaxis and disease treatment [53].

## 5. Conclusions

As for the core set for the assessment of the living environment of older people living in Poland, it is a comprehensive and important set of 20 ICF codes that reflect the most important elements of the environment affecting the health and functioning of the elderly. This set can contribute to the optimal management of care services and support in the area of adapting the environment to the older population. The core set for environmental assessment was developed for use by medical and care facilities, as well as by social workers, who should also pay attention to the elements of the environment that affect the level of functioning of older people. In the future, it may also form the basis of national surveys and screening tests for the assessment of the living environment of older people. Additionally, an abbreviated version of 10 codes was developed, containing mainly modifiable factors. This version is recommended for use by healthcare professionals, especially physicians, physical therapists, occupational therapists, and health visitors. Optimizing and enhancing the surrounding environment can contribute to a greater degree of independence, even with existing health problems in the older population.

## Figures and Tables

**Figure 1 ijerph-19-16341-f001:**
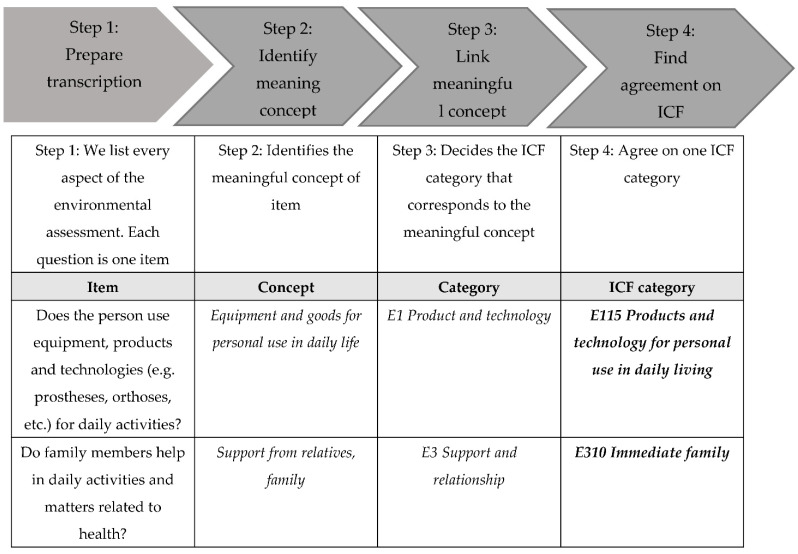
An illustration of the linking process.

**Table 1 ijerph-19-16341-t001:** The structure of the study group in the qualitative research.

Sociodemographic Features (*n* = 30)	Total
1. Age (mean, SD)	76.2 (3.7)
2. Gender *n* (%)	
Females	19 (63.33)
Males	11 (36.67)
3. Place of residence *n* (%)	
Town	15 (50.00)
Village	15 (50.00)
4. Marital status n (%)	
In a relationship	16 (53.33)
Single	14 (46.67)
5. Education *n* (%)	
At most vocational	18 (60.00)
At least secondary	12 (40.00)
6. Number of chronic diseases (mean, SD)	5.5 (3.1)

**Table 2 ijerph-19-16341-t002:** Selection of codes in the first four phases of the process.

No.	Code Number	Code Name	Phase 1 Review of the Literature	Phase 2 The Empirical Multicenter Study	Phase 3 The Elderly Focus Group	Phase 4 The Experts’ Study	Final Result Discussed at the Consensus Conference
		**Chapter 1 Products and Technology**					
1.	E110	Products or substances for personal consumption	x	x	x	x	x
2.	E115	Products and technology for personal use in daily living	x	x	x	x	x
3.	E120	Products and technology for personal indoor and outdoor mobility and transportation	x	x	x	x	x
4.	E125	Products and technology for communication	x	x	x	x	x
5.	E140	Products and technology for culture, recreation, and sport	-	x	-	-	-
6.	E145	Products and technology for the practice of religion and spirituality	x	-	-	-	-
7.	E150	Design, construction, and building products and technology of buildings for public use	x	x	x	x	x
8.	E155	Design, construction, and building products and technology of buildings for private use	x	x	x	x	x
9.	E160	Products and technology of land development	x	-	-	-	-
		**Chapter 2 Natural environment and human-made changes to the environment**					
10.	E210	Physical geography	x	-	x	x	x
11.	E215	Population	x	x	-	-	-
12.	E240	Light	x	x	x	x	x
13.	E245	Time-related changes	x	-	-	-	-
14.	E250	Sound	x	-	-	-	-
15.	E260	Air quality	x	x	x	x	x
		**Chapter 3 Support and relationships**					
16.	E310	Immediate family	x	x	x	x	x
17.	E315	Extended family	x	-	-	-	-
18.	E320	Friends	x	x	-	-	-
19.	E325	Acquaintances, peers, colleagues, neighbors, and community members	x	x	x	x	x
20.	E330	People in positions of authority	x	-	-	-	-
21.	E340	Personal care providers and personal assistants	x	x	x	x	x
22.	E355	Health professionals	x	x	x	x	x
23.	E360	Other professionals	x	-	-	-	-
		**Chapter 4 Attitudes**					
24.	E410	Individual attitudes of immediate family members	x	x	x	x	x
25.	E415	Individual attitudes of extended family members	x	-	-	-	-
26.	E420	Individual attitudes of friends	x	x	-	-	-
27.	E425	Individual attitudes of acquaintances, peers, colleagues, neighbors, and community members	x	x	x	x	x
28.	E430	Individual attitudes of people in positions of authority	x	-	-	-	-
29.	E440	Individual attitudes of personal care providers and personal assistants	x	x	x	x	x
30.	E450	Individual attitudes of health professionals	x	x	x	x	x
31.	E455	Individual attitudes of other professionals	x	-	-	-	-
32.	E460	Societal attitudes	-	x	-	x	-
33.	E465	Social norms, practices, and ideologies	x	-	-	-	-
		**Chapter 5 Services, systems, and policies**					
34.	E520	Open space planning services, systems, and policies	x	-	-	-	-
35.	E555	Associations and organizational services, systems, and policies	x	-	-	x	-
36.	E570	Social security services, systems, and policies	x	x	x	x	x
37.	E575	General social support services, systems, and policies	x	x	x	x	x
38.	E580	Health services, systems, and policies	x	x	x	x	x

**Table 3 ijerph-19-16341-t003:** The Polish final version of the simple, intuitive description of the ICF categories and example questions.

**Chapter 1 Products and Technology**				
**No.**	**Code Number**	**Code Name**	**Original Definition**	**Simple, Intuitive Descriptions and Example Questions**
1.	E110	Products or substances for personal consumption	Any natural or human-made object or substance gathered, processed, or manufactured for ingestion. Inclusions: food and drugs	Simplified definition: A natural or man-made product or substance for human consumption, including food, drugs, vitamins, and supplements. Sample interview question: Do you take medications and/or supplements regularly, and what is their importance for your health? (a) Facilitate: How much? + ………………. Qualifier (b) Neither facilitate nor create a barrier—meaningless 0 Qualifier (c) Create a barrier: How much? − ………………. Qualifier (d) 8—not specified (e) 9—not applicable Commentary on the answer given: …………………………………….
2.	E115	Products and technology for personal use in daily living	Equipment, products, and technologies used by people in daily activities, including those adapted, specially designed, or located in, on, or near the person using them. Inclusions: general and assistive products and technology for personal use	Simplified definition: Equipment, products, and technologies used by people in daily activities, including those located in, on, or near the person using them, e.g., prostheses, orthoses, or home equipment. Sample interview question: Do you use orthopaedic equipment and/or devices (e.g., prostheses, orthoses, etc.) for your daily activities, and what is their impact on your health and daily functioning? (a) Facilitate: How much? + ………………. Qualifier (b) Neither facilitate nor create a barrier—meaningless 0 Qualifier (c) Create a barrier: How much? − ………………. Qualifier (d) 8—not specified (e) 9—not applicable Commentary on the answer given: …………………………………….
3.	E120	Products and technology for personal indoor and outdoor mobility and transportation	Equipment, products, and technologies used by people in activities of moving inside and outside buildings, including those adapted, specially designed, or located in, on, or near the person using them. Inclusions: general and assistive products and technology for personal indoor and outdoor mobility and transportation	Simplified definition: Equipment, products, and technologies used by people to move inside and outside buildings, e.g., orthopaedic wheelchairs, cars. Sample interview question: Do you use devices to move inside (e.g., wheelchairs) and/or outside buildings (e.g., cars, vehicles) for your daily activities, and what is their impact on your health and daily functioning? (a) Facilitate: How much? + ………………. Qualifier (b) Neither facilitate nor create a barrier—meaningless 0 Qualifier (c) Create a barrier: How much? − ………………. Qualifier (d) 8—not specified (e) 9—not applicable Commentary on the answer given: …………………………………….
4.	E125	Products and technology for communication	Equipment, products and technologies used by people in activities of sending and receiving information, including those adapted, specially designed, or located in, on, or near the person using them. Inclusions: general and assistive products and technology for communication	Simplified definition: Equipment, products, and technology used by people to send and receive information, including those specially designed or located in, on, or near the person using them, such as optical and hearing aids and communication boards. Sample interview question: Do you use equipment, products, and technologies to send and receive information in your daily activities (e.g., optical and hearing aids, communication boards), and what is their impact on your health and daily functioning? (a) Facilitate: How much? + ………………. Qualifier (b) Neither facilitate nor create a barrier—meaningless 0 Qualifier (c) Create a barrier: How much? − ………………. Qualifier (d) 8—not specified (e) 9—not applicable Commentary on the answer given: …………………………………….
5.	E150	Design, construction, and building products and technology of buildings for public use	Products and technology that constitute an individual’s indoor and outdoor human-made environment that is planned, designed, and constructed for public use, including those adapted or specially designed. Inclusions: design, construction, and building products and technology of entrances and exits, facilities, and routing	Simplified definition: Products and technologies used to create appropriate conditions for people to move inside and outside buildings for public use, e.g., ramps, lifts. Sample interview question: How do you assess the functionality and adjustment of public places in your environment in the context of your health and current fitness needs? (a) Facilitate: How much? + ………………. Qualifier (b) Neither facilitate nor create a barrier—meaningless 0 Qualifier (c) Create a barrier: How much? − ………………. Qualifier (d) 8—not specified (e) 9—not applicable Commentary on the answer given: …………………………………….
6.	E155	Design, construction, and building products and technology of buildings for private use	Products and technology that constitute an individual’s indoor and outdoor human-made environment that is planned, designed, and constructed for private use, including those adapted or specially designed. Inclusions: design, construction, and building products and technology of entrances and exits, facilities, and routing	Simplified definition: Products and technologies used to create appropriate conditions for people to move inside and outside buildings for private use, e.g., ramps, lifts, railings, thresholds, bathroom amenities, adjusted kitchen cupboards. Sample interview question: How do you assess the functionality and adjustment of the interior of your place of residence in the context of your health and current fitness needs (e.g., height of worktop and cupboards, presence or absence of thresholds, presence of grips)? (a) Facilitate: How much? + ………………. Qualifier (b) Neither facilitate nor create a barrier—meaningless 0 Qualifier (c) Create a barrier: How much? − ………………. Qualifier (d) 8—not specified (e) 9—not applicable Commentary on the answer given: …………………………………….
**Chapter 2 Natural environment and human-made changes to the environment**				
**No.**	**Code number**	**Code name**	**Original definition**	**Simple, intuitive descriptions and example questions**
7.	E210	Physical geography	Features of land forms and bodies of water. Inclusions: features of geography included within orography (relief, quality, and expanse of land and land forms, including altitude) and hydrography (bodies of water such as lakes, rivers, seas)	Simplified definition: Features and forms of the terrain, e.g., elevation of the terrain. Sample interview question: How do you assess the terrain in the area where you live (e.g., flat, mountainous) in terms of your health and current fitness needs? (a) Facilitate: How much? + ………………. Qualifier (b) Neither facilitate nor create a barrier—meaningless 0 Qualifier (c) Create a barrier: How much? − ………………. Qualifier (d) 8—not specified (e) 9—not applicable Commentary on the answer given: …………………………………….
8.	E240	Light	Electromagnetic radiation by which things are made visible by either sunlight or artificial lighting (e.g., candles, oil or paraffin lamps, fires, and electricity) and which may provide useful or distracting information about the world. Inclusions: light intensity; light quality; color contrasts	Simplified definition: Light radiation, both solar and artificial lighting, providing useful or distracting information about the world. Sample interview question: How do you assess the lighting and color contrasts of your place of residence in the context of moving around and daily activities? (a) Facilitate: How much? + ………………. Qualifier (b) Neither facilitate nor create a barrier—meaningless 0 Qualifier (c) Create a barrier: How much? − ………………. Qualifier (d) 8—not specified (e) 9—not applicable Commentary on the answer given: …………………………………….
9.	E260	Air quality	Characteristics of the atmosphere (outside buildings) or enclosed areas of air (inside buildings), which may provide useful or distracting information about the world. Inclusions: indoor and outdoor air quality	Simplified definition: Air quality in the place of residence. Sample interview question: How do you assess the air quality in your place of residence in terms of its impact on your health and daily functioning? (a) Facilitate: How much? + ………………. Qualifier (b) Neither facilitate nor create a barrier—meaningless 0 Qualifier (c) Create a barrier: How much? − ………………. Qualifier (d) 8—not specified (e) 9—not applicable Commentary on the answer given: …………………………………….
**Chapter 3 Support and relationships**				
**No.**	**Code number**	**Code name**	**Original definition**	**Simple, intuitive descriptions and example questions**
10.	E310	Immediate family	Individuals related by birth, marriage, or other relationships recognized by the culture as immediate family, such as spouses, partners, parents, siblings, children, foster parents, adoptive parents, and grandparents. Exclusions: extended family (e315); personal care providers and personal assistants (e340)	Simplified definition: People related by birth, marriage, or other relationships recognized by the cultural norms as immediate family, such as: spouses, partners, parents, siblings, children, foster families, adoptive parents, and grandparents. Sample interview question: Do you receive help from your immediate family in daily activities, and what is its importance in the context of your health and daily functioning? (a) Facilitate: How much? + ………………. Qualifier (b) Neither facilitate nor create a barrier—meaningless 0 Qualifier (c) Create a barrier: How much? − ………………. Qualifier (d) 8—not specified (e) 9—not applicable Commentary on the answer given: …………………………………….
11.	E325	Acquaintances, peers, colleagues, neighbors, and community members	Individuals who are familiar with each other as acquaintances, peers, colleagues, neighbors, and community members in situations of work, school, recreation, or other aspects of life and who share demographic features such as age, gender, religious creed, or ethnicity or pursue common interests. Exclusions: associations and organizational services (e5550)	Simplified definition: Friends, colleagues, neighbors, and members of the local community. Sample interview question: Do you receive help from friends and/or neighbors in daily activities, and what is its importance in the context of your health and daily functioning? (a) Facilitate: How much? + ………………. Qualifier (b) Neither facilitate nor create a barrier—meaningless 0 Qualifier (c) Create a barrier: How much? − ………………. Qualifier (d) 8—not specified (e) 9—not applicable Commentary on the answer given: …………………………………….
12.	E340	Personal care providers and personal assistants	Individuals who provide services as required to support individuals in their daily activities and maintenance of performance in work, education, or other life situations, provided either through public or private funds or on a voluntary basis, such as providers of support for home-making and maintenance, personal assistants, transport assistants, paid help, nannies, and others who function as primary caregivers. Exclusions: immediate family (e310); extended family (e315); friends (e320); general social support services (e5750); health professionals (e355)	Simplified definition: People who provide services to help and support individuals in their daily activities or other life situations, paid for by both public and private funds or on a voluntary basis. Sample interview question: Do you receive help from a caregiver or a personal assistant in daily activities, and what is its importance in the context of your health and daily functioning? (a) Facilitate: How much? + ………………. Qualifier (b) Neither facilitate nor create a barrier—meaningless 0 Qualifier (c) Create a barrier: How much? − ………………. Qualifier (d) 8—not specified (e) 9—not applicable Commentary on the answer given: …………………………………….
13.	E355	Healthcare professionals (healthcare professionals)	Everyone who offers healthcare services, such as: doctors, nurses, midwives, physiotherapists, occupational therapists, speech therapists, audiologists, prosthetists, and medical social workers. Exclusion: other professionals (e360)	Simplified definition: Healthcare professionals, such as doctors, nurses, midwives, physiotherapists, occupational therapists, speech therapists, audiologists, prosthetists, and medical social workers. Sample interview question: How do you assess the quality of services provided by people working in healthcare (doctors, nurses, physiotherapists, etc.), and what is their importance in the context of your health? (a) Facilitate: How much? + ………………. Qualifier (b) Neither facilitate nor create a barrier—meaningless 0 Qualifier (c) Create a barrier: How much? − ………………. Qualifier (d) 8—not specified (e) 9—not applicable Commentary on the answer given: …………………………………….
**Chapter 4 Attitudes**				
**No.**	**Code number**	**Code name**	**Original definition**	**Simple, intuitive descriptions and example questions**
14.	E410	Individual attitudes of immediate family members	General or specific opinions and beliefs of immediate family members about the person or about other matters (e.g., social, political, and economic issues) that influence individual behavior and actions.	Simplified definition: Opinions and beliefs of immediate family members that affect individual behavior and actions. Sample interview question: Do family opinions and beliefs regarding people and/or social, political, and economic issues have an impact on providing you with help and support in activities related to your health and daily functioning? (a) Facilitate: How much? + ………………. Qualifier (b) Neither facilitate nor create a barrier—meaningless 0 Qualifier (c) Create a barrier: How much? − ………………. Qualifier (d) 8—not specified (e) 9—not applicable Commentary on the answer given: …………………………………….
15.	E425	Individual attitudes of acquaintances, peers, colleagues, neighbors, and community members	General or specific opinions and beliefs of acquaintances, peers, colleagues, neighbors, and community members about the person or about other matters (e.g., social, political, and economic issues) that influence individual behavior and actions.	Simplified definition: Opinions and beliefs of friends, colleagues, neighbors, and members of the local community that affect individual behavior and actions. Sample interview question: Do the opinions and beliefs of friends, peers, colleagues, neighbors, and members of the local community regarding people and/or social, political, and economic issues have an impact on providing you with help and support in activities related to your health and daily functioning? (a) Facilitate: How much? + ………………. Qualifier (b) Neither facilitate nor create a barrier—meaningless 0 Qualifier (c) Create a barrier: How much? − ………………. Qualifier (d) 8—not specified (e) 9—not applicable Commentary on the answer given: …………………………………….
16.	E440	Individual attitudes of personal care providers and personal assistants	General or specific opinions and beliefs of personal care providers and personal assistants about the person or about other matters (e.g., social, political, and economic issues) that influence individual behavior and actions.	Simplified definition: Opinions and beliefs of caregivers and personal assistants that affect individual behavior and actions. Sample interview question: Do the opinions and beliefs of the caregiver/personal assistant regarding people and/or social, political, and economic issues have an impact on providing you with help and support in activities related to your health and daily functioning? (a) Facilitate: How much? + ………………. Qualifier (b) Neither facilitate nor create a barrier—meaningless 0 Qualifier (c) Create a barrier: How much? − ………………. Qualifier (d) 8—not specified (e) 9—not applicable Commentary on the answer given: …………………………………….
17.	E450	Individual attitudes of health professionals	General or specific opinions and beliefs of health professionals about the person or about other matters (e.g., social, political, and economic issues) that influence individual behavior and actions.	Simplified definition: Opinions and beliefs of healthcare professionals that affect individual behavior and actions. Sample interview question: Do the opinions and beliefs of healthcare professionals regarding people and/or social, political, and economic issues have an impact on providing you with help and support in activities related to your health and daily functioning? (a) Facilitate: How much? + ………………. Qualifier (b) Neither facilitate nor create a barrier—meaningless 0 Qualifier (c) Create a barrier: How much? − ………………. Qualifier (d) 8—not specified (e) 9—not applicable Commentary on the answer given: …………………………………….
**Chapter 5 Services, systems, and policies**				
**No.**	**Code number**	**Code name**	**Original definition**	**Simple, intuitive descriptions and example questions**
18.	E570	Social security services, systems, and policies	Services, systems, and policies aimed at providing income support to people who, because of age, poverty, unemployment, health condition, or disability, require public assistance that is funded either by general tax revenues or contributory schemes. Exclusion: economic services, systems, and policies (e565)	Simplified definition: Services, systems, and policies aimed at providing income support to people who, because of their age, poverty, unemployment, health, or disability, require public assistance that is funded either by general tax revenues or contributory schemes. Sample interview question: How do you assess the social security support system in your place of residence/country in the context of your health needs? (a) Facilitate: How much? + ………………. Qualifier (b) Neither facilitate nor create a barrier—meaningless 0 Qualifier (c) Create a barrier: How much? − ………………. Qualifier (d) 8—not specified (e) 9—not applicable Commentary on the answer given: …………………………………….
19.	E575	General social support services, systems, and policies	Services, systems, and policies aimed at providing support to those requiring assistance in areas such as shopping, housework, transport, self-care, and care of others in order to function more fully in society. Exclusions: social security services, systems, and policies (e570); personal care providers and personal assistants (e340); health services, systems, and policies (e580)	Simplified definition: Services, systems, and policies aimed at providing support to those requiring assistance in areas such as shopping, housework, transport, self-care, and care of others in order to function more fully in society. Sample interview question: How do you assess the social security support system in your place of residence/country in the context of your needs in daily activities necessary for the fullest functioning in society? (a) Facilitate: How much? + ………………. Qualifier (b) Neither facilitate nor create a barrier—meaningless 0 Qualifier (c) Create a barrier: How much? − ………………. Qualifier (d) 8—not specified (e) 9—not applicable Commentary on the answer given: …………………………………….
20.	E580	Health services, systems, and policies	Services, systems, and policies for preventing and treating health problems, providing medical rehabilitation, and promoting a healthy lifestyle. Exclusion: general social support services, systems, and policies (e575)	Simplified definition: Services, systems, and policies for preventing and treating health problems, providing medical rehabilitation, and promoting a healthy lifestyle. Sample interview question: How do you assess the health support system in your place of residence/country in the context of your health needs? (a) Facilitate: How much? + ………………. Qualifier (b) Neither facilitate nor create a barrier—meaningless 0 Qualifier (c) Create a barrier: How much? − ………………. Qualifier (d) 8—not specified (e) 9—not applicable Commentary on the answer given: …………………………………….

## Data Availability

The datasets used and analyzed in the current study are available from the corresponding author on reasonable request.

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
