# Peer review of "Development of the Polish Version of the ICF Core Set for the Environment of Older People"

_ijerph, 2022, doi:10.3390/ijerph192316341_

Round 1
Reviewer 1 Report
Dear authors, I wanted to make some recommendations that can improve your manuscript.
Materials and methods
1. First phase. What were the inclusion and exclusion criteria for the selection of studies? What was the search strategy used? Was the PRISMA checklist applied?
2. Second phase. What criteria were applied for the selection of the 10 centers? Was random sampling used for the selection of centers?
Author Response
Dear Reviewer,
Thank you very much for your re-reviews and additional tips how to improve our manuscript.
The manuscript has been revised according to your suggestions. Changes made to the main manuscript are marked in green. We also provide answers to individual comments below.
We hope that you will be satisfied with the change made and agree to publish our article.
Yours sincerely,
Agnieszka Ćwirlej-Sozańska
Rev. 1
- First phase. What were the inclusion and exclusion criteria for the selection of studies? What was the search strategy used? Was the PRISMA checklist applied?
Response:
Thank you for your attention. The systematic review of the literature as one of the stages of the preparation of the ICF Core Set was prepared in accordance with the methodology described by Selb et al. in "A guide on how to develop an International Classification of Functioning, Disability and Health Core Set [1]. To some extent, it coincides with the PRISMA guidelines. However, it gives the researcher more freedom to choose articles.
Selb M, Escorpizo R, Kostanjsek N, Stucki G, Üstün B, Cieza A. A guide on how to develop an International Classification of Functioning, Disability and Health Core Set. Eur J Phys Rehabil Med. 2015;51(1):105-17.
The paragraph describing how to conduct a systematic literature review in the Material and Method section is presented as follows:
“First phase was the systematic review of the literature to identify environmental factors posing obstacles and barriers as well as to facilitate the functioning of older people aged 65 and over. The study was conducted in accordance with the methodology indicated by the WHO ICF Research Branch [20]. The following databases were searched: PubMed, MEDLINE, Google Scholar, and ISI Web of Knowledge. The following MeSH headings were used: Aged, 80 and over, Activities of Daily Living, Walking, Residence Characteristics, Environment. Qualification of publications was based on an analysis of the title, abstract, and then the full-text version. The following issues were taken into account: the reviewed articles in English, articles assessing the living environment of the elderly, randomized controlled trials, clinical controlled trials, cross-sectional studies, observational studies, and qualitative studies. Searching for selected MeSH in medical databases provided 2988 records, and subsequently removing duplicates and items not related to the subject after an initial analysis of abstracts, a total of 534 references remained. Then, publications that did not have a full text were removed, resulting in 517 articles. Ultimately, 97 articles were included in the analysis, which were used to collect the concepts of the positive and negative impact of the environment on functioning and disability of the elderly. The issues gathered from the literature review were linked with ICF categories using standard linking rules [21]. An illustration of how concepts are related to ICF categories is shown in Figure 1.”
- Second phase. What criteria were applied for the selection of the 10 centers? Was random sampling used for the selection of centers?
Response:
Thank you for your questions.
The criteria for the inclusion of the centers were: at least 3 years of operation, profile of medical, social, caring, assistance or mixed activity addressed to the elderly, consent and readiness of the center to participate in the study.
In the first stage, a database of centers and organizations cooperating or caring for the elderly was collected. The second stage was to randomly select 5 medical centers and 5 mixed centers or centers supporting the elderly from this database. If a given center refused to cooperate or did not have an expert with at least 3 years of work experience, another center was drawn from the pool. We have detailed the record in the article.
The description of the empirical multicenter study in the manuscript is detailed as follows:
„Second phase was the empirical multicenter study, which aimed to identify the problems experienced by older adults as a result of environmental barriers that are documented in institutional settings. In the first stage of this study, a database of centers and organizations cooperating or caring for the elderly was collected. The inclusion criteria for the center were at least 3 years of operation in the market, a profile of medical, social, caring, supporting or mixed activities addressed to the older people, consent of the center and willingness to participate in the study. The second stage was the selection of 5 medical centers and 5 centers of mixed or supporting profiles for the elderly from the gathered database. If a given center refused to cooperate or did not have an expert with at least 3 years of work experience, another center was drawn from the pool.
In the course of the empirical multicenter study, employees of the institution participating in the study conducted their standard measurements and examinations with the beneficiaries, extended by a deepen, semi-structured interview focusing on problems in daily functioning in the context of the environment and living conditions of the subjects. The instrument used to collect the data was the ICF checklist enriched with environmental codes selected after a systematic review of the literature. The ICF checklist required the researchers to evaluate the extent of the problem in each ICF category and the size of the problem / facilitator in terms of environmental factors. In addition, it was asked to focus in particular on the relationship between the categories of functioning and activities with environmental categories and to mark those that had a substantial impact on the improvement or deterioration of the performance of activities. A category that has been identified as a problem, barrier and / or facilitator for minimum 25% of the subjects were included in the list of candidate categories.”
Additionally, the manuscript was proofread for linguistic errors.

Reviewer 2 Report
This is an interesting paper, well-written and an important topic. However, it's not totally clear why you would not just use the full ICF list when assessing older people?
Author Response
Dear Reviewer,
Thank you very much for your re-reviews and additional tips how to improve our manuscript.
The manuscript has been revised according to your suggestions. Changes made to the main manuscript are marked in green. We also provide answers to individual comments below.
We hope that you will be satisfied with the change made and agree to publish our article.
Yours sincerely,
Agnieszka Ćwirlej-Sozańska
Rev 2.
- This is an interesting paper, well-written and an important topic. However, it's not totally clear why you would not just use the full ICF list when assessing older people?
Response:
Thank you very much for your attention. Based on many years of experience in research and implementation of ICF in Poland, we noticed the lack of an appropriate tool to assess the living environment of older people living in the community. Therefore, we decided to develop the Core Set as a response to the missing tool that would complement the assessment of activity and participation of older people with the environmental context. Ultimately, our work aims to create a uniform tool based on ICF (similarly to WHODAS 2.0 built on the basis of selected ICF activity codes).
Additionally, the manuscript was proofread for linguistic errors.

Reviewer 3 Report
ICF conceptualizes the level of functioning as a dynamic interaction between an individual's health status, environmental factors, and personal factors, which is seen as a good attempt in terms of developing a biopsycholsocial model of the elderly and the environment.
However, many ICF studies have already been produced, and this study needs to be more reader-friendly. As a reader, I pointed out some problem.
1. Please schematically illustrate the link between the reviewed literature and the ICF so that the reader can easily understand the reference review
2. A part of this study is a qualitative study and it was conducted. So, please express the process and results of the qualitative study in detail. Please refer to the criteria of COREQ. Please pay attention to readability as well.
3. Can you express the baseline characteristics of qualitative research participants in the form of a table?
4. Please describe the contribution made by an expert in a certain field. This attempt may only be the first step in developing standardized tools. Therefore, cooperation between various experts is essential and its contents need to be reported in manuscript.
5. Please express the percentage of experiments will to include the named category in the ICF Core Set in the form of a graph
6. Was there any overlap or major difference between the four different results?
7. There is red letters in Author Contributions. Please correct this.
Author Response
Dear Reviewer,
Thank you very much for your feedback and additional tips on how to improve our manuscript.
The manuscript has been revised according to your suggestions. Changes to the main manuscript are marked in green. We also provide answers to individual comments in the appendix.
We hope that you will be satisfied with the change and agree to publish our article.
Yours sincerely,
Agnieszka Ćwirlej-Sozańska

Round 2
Reviewer 1 Report
Thanks to the authors for addressing and resolving all recommended comments
Reviewer 3 Report
1. Please describe the methods you used for authenticity and trustworthiness in qualitative research.
Ex) triangulation.
2. Please mention which of the various approaches to qualitative research you have used.
Ex) case study...
3.If two or more different research methods were used, it is appropriate to view it as a mixed method study. If so, please describe which of the various combination methods of the Mixed Method you used.
Ex) nesting
These are the contents that must be included even if it is assumed that qualitative research is written as an aid.
Author Response
Dear Reviewer,
Thank you very much for your reviews and additional tips how to improve our manuscript.
The manuscript has been revised according to your suggestions. Changes made to the main manuscript are marked in red We also provide answers to individual comments below.
We hope that you will be satisfied with the change made and agree to publish our article.
Yours sincerely,
Agnieszka Ćwirlej-Sozańska
- Please describe the methods you used for authenticity and trustworthiness in qualitative research.
Ex) triangulation.
Response:
Thank you for your very important attention. We have added the following supplement to our manuscript:
“The qualitative study was based on the method of triangulation of research methods using semiotics, in-depth individual interviews and focus group interviews in the research project. This made it possible to obtain a broader research material and a more complete basis for its interpretation.”
- Please mention which of the various approaches to qualitative research you have used.
Ex) case study...
Response:
Thank you for your very important attention. The study used research methods such as in-depth interviews and the focus group method. Included in the description above.
- If two or more different research methods were used, it is appropriate to view it as a mixed method study. If so, please describe which of the various combination methods of the Mixed Method you used.
Ex) nesting
Response:
The entire study (5 of its stages) can be considered as a method of multi-level analysis (nesting). In the qualitative study itself, in which the elderly took part, not completely, because the elderly were interviewed individually or in a focus group. The following addition has been added to the manuscript:
“The tool preparation process was carried out in multi-level analysis in 5 stages:”